# Is Your LLM Overcharging You?
## Tokenization, Transparency, and Incentives

Ander Artola Velasco [1]   Stratis Tsirtsis [1]   Nastaran Okati [1]   Manuel Gomez-Rodriguez [1]

## Abstract

State-of-the-art large language models require specialized hardware and substantial energy to operate. As a consequence, cloud-based services that provide access to large language models have become very popular. In these services, the price users pay for an output provided by a model depends on the number of tokens the model uses to generate it—they pay a fixed price per token. In this work, we show that this pricing mechanism creates a financial incentive for providers to strategize and misreport the (number of) tokens a model used to generate an output, and users cannot prove, or even know, whether a provider is overcharging them. However, we also show that, if an unfaithful provider is obliged to be transparent about the generative process used by the model, misreporting optimally without raising suspicion is hard. Nevertheless, as a proof-of-concept, we introduce an efficient heuristic algorithm that allows providers to significantly overcharge users without raising suspicion, highlighting the vulnerability of users under the current pay-per-token pricing mechanism. Further, to completely eliminate the financial incentive to strategize, we introduce a simple incentive-compatible token pricing mechanism. Under this mechanism, the price users pay for an output provided by a model depends on the number of characters of the output—they pay a fixed price per character. Along the way, to illustrate and complement our theoretical results, we conduct experiments with several large language models from the `Llama`, `Gemma` and `Ministral` families, and input prompts from the LMSYS Chatbot Arena platform.

[1]Max Planck Institute for Software Systems, Kaiserslautern, Germany. Correspondence to: Ander Artola Velasco <avelasco@mpi-sws.org>.

*Non-archival presentation at ICML 2025 Tokenization Workshop (TokShop)*, Vancouver, Canada. 2025.

## 1. Introduction

Large language models (LLMs) are becoming ubiquitous across multiple industries—from powering chatbots and virtual assistants to driving innovation in research, healthcare, and finance (Bubeck et al., 2023; Mozannar et al., 2024; Haupt & Marks, 2023; Romera-Paredes et al., 2024). However, since the computational resources required to run these models are significant, most (enterprise) users are unable to host them locally. As a result, users rely on a few cloud-based providers that offer LLMs-as-a-service to obtain access (Chen et al., 2023; Snell et al., 2024; Pais et al., 2022; Patel et al., 2024).

In a typical LLM-as-a-service, a user submits a prompt to the provider via an application programming interface (API). Then, the provider feeds the prompt into an LLM running on their own hardware, which (stochastically) generates a sequence of tokens as an output using a generative process.[1] Finally, the provider shares the output with the user and charges them based on a simple pricing mechanism: a fixed price per token.[2] In this paper, we focus on the following fundamental question:

*What incentives does the pay-per-token pricing mechanism create for providers?*

Our key observation is that, in the interaction between a user and a provider, there is an asymmetry of information (Milgrom & Roberts, 1987; Rasmusen, 1989; Mishra et al., 1998). The provider observes the entire generative process used by the model to generate an output, including its intermediate steps and the final output tokens, whereas the user only observes and pays for the (output) tokens shared with them by the provider. This asymmetry sets the stage for a situation known in economics as moral hazard (Holmström, 1979), where one party (the provider) has the opportunity to take actions that are not observable by the other party (the user) to maximize their own utility at the expense of the other party.

---

[1]Tokens are units that make up sentences and paragraphs, such as (sub-)words, symbols and numbers.

[2]https://ai.google.dev/gemini-api/docs/pricing, https://openai.com/api/pricing.

The core of the problem lies in the fact that the tokenization of a string is not unique. For example, consider that the user submits the prompt "`Where does the next NeurIPS take place?`" to the provider, the provider feeds it into an LLM, and the model generates the output "`|San| Diego|`" consisting of two tokens. Since the user is oblivious to the generative process, a self-serving provider has the capacity to misreport the tokenization of the output to the user without even changing the underlying string. For instance, the provider could simply share the tokenization "`|S|a|n| |D|i|e|g|o|`" and overcharge the user for nine tokens instead of two!

A simple remedy to build trust between the two parties would be to require providers to share with the user more information about the generative process used by the model, such as the next-token distribution in each step of the process. This would grant the user a form of (partial) auditability, since tokenizations, such as the one mentioned above, may have negligible probability in practice. Importantly, if the provider implements procedures to prevent the generation of low-probability tokens (*e.g.*, top-$p$ sampling (Holtzman et al., 2019), top-$k$ sampling), as commonly done in practice, such tokenizations would not only be unlikely, but rather implausible, giving grounds to the user to contest the specific tokenization of the output shared with them by the provider. In this case, a provider would have to invest additional effort (and resources) to misreport the tokenization of an output while preserving its plausibility, making such a strategic behavior significantly less worthy from a financial point of view.

However, some providers may be highly reluctant to share information that could potentially expose the internal workings of their LLMs, especially if the LLMs are proprietary and such information can be used by competitors (Carlini et al., 2024). In the absence of any additional means for the users to verify the truthfulness of the providers, the only remaining option is to regulate the transactions between users and providers in a way that eliminates the incentive for providers to engage in misreporting in the first place. To this end, we introduce and argue for a pay-per-character pricing mechanism that serves exactly this purpose.

**Our contributions.** We start by characterizing tokenization (mis)reporting in LLMs as a principal-agent problem (Grossman & Hart, 1992; Bolton & Dewatripont, 2004; Dütting et al., 2024). Building upon this characterization, we make the following contributions:

1. We show that, under the pay-per-token pricing mechanism, providers have a financial incentive to (mis-)report each character of the outputs generated by the LLMs they serve as a separate token.

2. We show that, if the providers are transparent about the next-token distribution used by the LLMs they serve, they cannot expect to find the longest tokenization of an output that is plausible in polynomial time.

3. We introduce a heuristic algorithm that, as a proof-of-concept, allows providers to find plausible token sequences that are longer or equal than a generated output token sequence very efficiently.

4. We show that any incentive-compatible pricing mechanism must price tokens linearly on their character count. Moreover, we further show that, if each character is priced equally, there is only one incentive-compatible pricing mechanism, which we refer to as the pay-per-character pricing mechanism.

Along the way, to illustrate and complement the above contributions, we conduct a series of experiments using LLMs from the `Llama`, `Gemma` and `Ministral` families and user input prompts from the LMSYS Chatbot Arena platform.[3] Under the pay-per-token pricing mechanism, we empirically demonstrate that an unfaithful provider who is transparent about the generative process used by the LLM they serve can use our heuristic algorithm to overcharge users by up to ~13%.

**Further related work.** Our work builds upon further related work on tokenization, economics of LLMs-as-a-service, mechanism design, and game theory in LLMs.

Multiple lines of empirical evidence have shown that tokenization plays a central role in developing and analyzing LLMs (Rajaraman et al., 2024; Geh et al., 2024; Singh & Strouse, 2024; Giulianelli et al., 2024; Geh et al., 2025; Petrov et al., 2023; Ovalle et al., 2024; Chatzi et al., 2025; Benz et al., 2025). Consequently, there have been a variety of efforts focusing on better understanding and improving byte-pair encoding (BPE), the tokenization algorithm most commonly used in LLMs (Bostrom & Durrett, 2020; Kozma & Voderholzer, 2024; Zouhar et al., 2023; Lian et al., 2024b; Sennrich et al., 2016; Lian et al., 2024a). However, this line of work has overlooked the economic implications of tokenization (in the context of LLMs-as-a-service), which is the main focus of our work.

The literature on the economics of LLMs-as-a-service has been recently growing very rapidly (La Malfa et al., 2024; Bergemann et al., 2025; Mahmood, 2024; Laufer et al., 2024; Cai et al., 2025; Saig et al., 2024).

Similarly as in our work, Cai et al. (Cai et al., 2025) and Saig et al. (Saig et al., 2024) also study a setting in which the provider has a financial incentive to be unfaithful to the

---

[3]The code we used in our experiments is available at https://github.com/Networks-Learning/token-pricing.

users. However, in their setting, the provider has an incentive to be unfaithful about the LLM they use to generate outputs rather than the tokenization of the outputs—it may use a cheaper-to-run LLM than the one it charges the users for. To reduce the financial incentive to strategize, Cai et al. (Cai et al., 2025) argue for solutions based on increased transparency as well as trusted execution environments, and Saig et al. (Saig et al., 2024) argue for a pay-for-performance pricing mechanism using a contract theory formulation. The work most similar to ours is that of Sun et al. (Sun et al., 2025a), who study the problem of provider reporting artificially injected tokens during the hidden reasoning process used in the recent test-time compute approach to inference. However, they limit their analysis to injecting additional reasoning tokens and do not consider retokenization of the output string, which is the main focus of our work. The literature on mechanism design and game theory in LLMs has explored incentive auction mechanisms for generated content (Duetting et al., 2024), LLM-augmented voting processes (Fish et al., 2023), and the potential of LLMs as economic agents (Horton, 2023; Raman et al., 2024; Zhang et al., 2024; Sun et al., 2025b; Kovarik et al., 2023). However, to the best of our knowledge, our work is the first to explore incentive-compatible token pricing mechanisms in LLMs.

## 2. A Principal-Agent Model of Delegated Autoregressive Generation

We characterize the interaction between a user and an LLM provider as a principal-agent problem (Grossman & Hart, 1992; Bolton & Dewatripont, 2004; Dütting et al., 2024), where the principal (the user) delegates a task (a generation) to the agent (the provider), who performs the task on behalf of the principal and gets paid based on a commonly agreed-upon contract.

In a typical interaction between a user and a provider, the user first submits a prompt $q \in \Sigma^*$ to the provider, where $\Sigma^*$ denotes the set of all finite-length strings over an alphabet (*i.e.*, a finite set of characters) $\Sigma$. Then, the provider uses their own hardware to query an LLM with the prompt $q$, and the LLM (stochastically) generates an output token sequence $\mathbf{t} = (t_1, t_2, \ldots, t_k) \in \mathcal{V}^*$ in an autoregressive manner, one token at a time. Here, $t_i \in \mathcal{V}$ is the $i$-th token in a sequence of $k$ tokens, $\mathcal{V} \subset \Sigma^*$ is the (token) vocabulary used by the LLM,[4] and $\mathcal{V}^*$ denotes the set of all finite-

length sequences over the vocabulary.[5] Finally, the provider reports to the user the generated output token sequence. Importantly, since the user is oblivious to the autoregressive process used by the LLM, the provider has the capacity to misreport the output token sequence to the user—the reported output token sequence $\tilde{\mathbf{t}}$ may not correspond to the generated output token sequence $\mathbf{t}$.

Before the interaction between a user and an LLM provider begins, both parties agree on a contract that specifies how the provider should be compensated for the output token sequence they report to the user. More specifically, the user and the provider agree on a *pricing mechanism* that determines the monetary reward $r\left(\tilde{\mathbf{t}}\right)$ that the user should transfer to the provider for the reported output token sequence $\tilde{\mathbf{t}}$:

**Definition 2.1** (Pricing mechanism). Given a vocabulary of tokens $\mathcal{V}$, a pricing mechanism is a function $r \colon \mathcal{V}^* \to \mathbb{R}_{\geq 0}$ that assigns a price to each reported output token sequence $\tilde{\mathbf{t}} \in \mathcal{V}^*$.

Throughout the paper, we focus on additive pricing mechanisms, which include the widely used pay-per-token pricing mechanism. An additive pricing mechanism independently assigns a price $r\left(\tilde{t}_i\right)$ to each token $\tilde{t}_i$ in a reported output token sequence $\tilde{\mathbf{t}}$, and calculates the price $r\left(\tilde{\mathbf{t}}\right)$ of a reported output token sequence by adding up the price of each individual token.

Given a generated output token sequence $\mathbf{t}$ and a reported output token sequence $\tilde{\mathbf{t}}$, the provider's utility $U_{\text{provider}}\left(\tilde{\mathbf{t}}, \mathbf{t}\right)$ is given by the difference between the monetary reward $r\left(\tilde{\mathbf{t}}\right)$ the provider receives from the user for $\tilde{\mathbf{t}}$ and the cost $c(\mathbf{t})$ of generating the output token sequence $\mathbf{t}$, *i.e.*,

$$U_{\text{provider}}\left(\tilde{\mathbf{t}}, \mathbf{t}\right) = r\left(\tilde{\mathbf{t}}\right) - c(\mathbf{t}). \tag{1}$$

Here, motivated by recent empirical studies showing that the energy consumption scales linearly with output length (Adamska et al., 2025; Fernandez et al., 2025), we assume that the cost of generating $\mathbf{t}$ is a linear function of its length, that is, $c(\mathbf{t}) = c_0 \cdot \texttt{len}(\mathbf{t})$, where $c_0 \in \mathbb{R}_{>0}$ is a constant that represents the running costs of generating a single token (*e.g.*, electricity costs, hardware maintenance), and $\texttt{len}(\mathbf{t})$ denotes the length (*i.e.*, number of tokens) of $\mathbf{t}$.

Given a reported output token sequence $\tilde{\mathbf{t}}$, the user's utility $U_{\text{user}}\left(\tilde{\mathbf{t}}\right)$ is given by the difference between the value $v(\tilde{\mathbf{t}})$ they derive from the sequence $\tilde{\mathbf{t}}$ and the monetary reward $r(\tilde{\mathbf{t}})$ they pay to the provider for $\tilde{\mathbf{t}}$, that is, $U_{\text{user}}\left(\tilde{\mathbf{t}}\right) = v\left(\tilde{\mathbf{t}}\right) - r\left(\tilde{\mathbf{t}}\right)$. However, the user typically derives value from the text that the output token sequence represents, rather than the token sequence itself. For exam-

---

[4]We assume $\Sigma \subset \mathcal{V}$ since this condition must occur for the vocabulary to be able to tokenize single characters. In this context, note that standard vocabulary-building algorithms such as BPE satisfy this by construction (Sennrich et al., 2016).

[5]In practice, the provider turns the prompt $q$ into a sequence of tokens using a tokenizer before passing it as input to the model, but modeling this explicitly is not relevant in our work.

ple, in creative writing, the user may be interested in the extent to which the generated text is captivating to read, and in code generation, the user may be interested in operational aspects of the generated code, such as its correctness and efficiency. Therefore, we assume that $v\left(\tilde{\mathbf{t}}\right) = v\left(\mathtt{str}(\tilde{\mathbf{t}})\right)$, where $\mathtt{str}\colon \mathcal{V}^* \to \Sigma^*$ maps a sequence of tokens to the respective string, and we use $|\mathtt{str}(\tilde{\mathbf{t}})|$ to denote the number of characters in the string $\mathtt{str}(\tilde{\mathbf{t}})$.

While the provider can, in principle, report any token sequence $\tilde{\mathbf{t}}$ they prefer (*e.g.*, the one that maximizes their reward based on the pricing mechanism), arbitrary manipulations of the generated output may easily raise suspicion about the provider's practices. Therefore, in our work, we restrict our focus to a more subtle strategy: misreporting the tokenization of the generated output sequence while preserving its string-level representation. Under this strategy, given a generated output token sequence $\mathbf{t}$ with $s = \mathtt{str}(\mathbf{t})$, the provider reports a token sequence $\tilde{\mathbf{t}}$ from the set $\mathcal{V}_s^* = \left\{\tilde{\mathbf{t}} \in \mathcal{V}^* \colon \mathtt{str}\left(\tilde{\mathbf{t}}\right) = s\right\}$. Then, it is easy to see that, as long as there exists a token sequence $\tilde{\mathbf{t}} \in \mathcal{V}_s^*$ such that $r\left(\tilde{\mathbf{t}}\right) > r\left(\mathbf{t}\right)$, it holds that

$$U_{\mathrm{provider}}\left(\tilde{\mathbf{t}}, \mathbf{t}\right) > U_{\mathrm{provider}}\left(\mathbf{t}, \mathbf{t}\right) \quad \text{and} \quad v\left(\tilde{\mathbf{t}}\right) = v(\mathbf{t}).$$

In other words, the provider has an incentive not to be truthful and potentially overcharge the user, and can do so in a way that maintains the value the user derives from the reported output sequence. In what follows, we will explore the conditions under which such strategic behavior can occur and remain undetected by the user. Later on, we will propose a pay-per-character pricing mechanism that provably eliminates the provider's incentive for this type of strategic behavior.

## 3. Provider Incentives under the Pay-Per-Token Pricing Mechanism

In this section, we analyze the pay-per-token-pricing mechanism using the principal-agent model introduced in Section 2. First, we show that, under this mechanism, the provider's utility is tightly linked to the length of the reported output token sequence—the longer the reported sequence, the higher the provider's utility. Then, we further show that, if the provider is required to be transparent about the next-token distribution used by the LLM they serve, they cannot expect to find the longest tokenization of a given output that appears to be plausible in polynomial time. Finally, we demonstrate that, in practice, this computational hardness does not preclude the provider from efficiently finding plausible tokenizations of a given output that increase its utility.

### 3.1. Pay-Per-Token Incentivizes (Mis-)Reporting Longer Tokenizations

To be profitable, a cloud-based LLM provider needs to at least amortize the cost of output generation. Therefore, under the assumption that the cost of output generation is a linear function of the output length, the widely used pay-per-token pricing mechanism is a natural choice.

**Definition 3.1** (Pay-per-token)**.** A pricing mechanism $r\colon \mathcal{V}^* \to \mathbb{R}_{\geq 0}$ is called pay-per-token if and only if it is additive and, for all $t \in \mathcal{V}$, it satisfies that $r(t) = r_0$, where $r_0 \geq 0$ is a constant price per token.

As an immediate consequence, under the pay-per-token pricing mechanism, the monetary reward that the provider receives from reporting an output token sequence $\tilde{\mathbf{t}}$ is a linear function of the output length, *i.e.*, $r\left(\tilde{\mathbf{t}}\right) = r_0 \cdot \mathtt{len}\left(\tilde{\mathbf{t}}\right)$. Further, since the cost to generate the output sequence $\mathbf{t}$ is independent of the reported output sequence $\tilde{\mathbf{t}}$, the provider's utility, given by Eq. 1, is simply a (linearly) increasing function of the length of the reported output sequence. That is, for any true output sequence $\mathbf{t}$ with $\mathtt{str}(\mathbf{t}) = s$, it holds that

$$U_{\mathrm{provider}}\left(\tilde{\mathbf{t}}, \mathbf{t}\right) > U_{\mathrm{provider}}\left(\tilde{\mathbf{t}}', \mathbf{t}\right) \quad \text{for any} \quad \tilde{\mathbf{t}}, \tilde{\mathbf{t}}' \in \mathcal{V}_s^*$$
$$\text{such that} \quad \mathtt{len}\left(\tilde{\mathbf{t}}\right) > \mathtt{len}\left(\tilde{\mathbf{t}}'\right). \quad (2)$$

Therefore, a rational provider seeking to maximize their utility needs to find a tokenization of $s$ with maximum length, *i.e.*,

$$\tilde{\mathbf{t}}_{\mathrm{max}} = \underset{\tilde{\mathbf{t}} \in \mathcal{V}_s^*}{\mathrm{argmax}} \; \mathtt{len}\left(\tilde{\mathbf{t}}\right). \quad (3)$$

Since LLM vocabularies typically include tokens corresponding to all individual characters (*i.e.*, $\Sigma \subset \mathcal{V}$), it is easy to see that the optimization problem admits a trivial solution: report each character in $s$ as a separate token. Strikingly, the financial incentive for (mis-)reporting this tokenization can be very significant in practice. For example, for input prompts from the LMSYS Chatbot Arena platform (Zheng et al., 2024), an unfaithful provider following such a strategy may overcharge users by $\sim 3\times$, as shown in Table 1 (refer to Appendix A for additional details regarding our experiments). Importantly, the user has no grounds to verify whether such a tokenization is indeed the one generated by the model, or if it has been manipulated by the provider. That being said, such tokenizations may arguably raise suspicion, particularly if the provider is required to be transparent about the next-token distribution used by the LLM they serve. Next, we will show that an unfaithful provider who aims to find the longest tokenization that maximizes their utility and appears to be plausible is likely to fail.

*Table 1.* **Financial gain from (mis-)reporting each output character as a separate token.** The results show the percentage of tokens overcharged by an unfaithful provider who (mis-)reports each character in the output token sequences generated by an LLM to 400 prompts from the LMSYS Chatbot Arena platform as a separate token. Here, we set the temperature of the model to 1.0 and repeat each experiment 5 times to obtain 90% confidence intervals.

| LLM | Overcharged tokens (%) |
| --- | --- |
| Llama-3.2-1B-Instruct | $344.9 \pm 3.8$ |
| Llama-3.2-3B-Instruct | $345.2 \pm 6.0$ |
| Gemma-3-1B-In | $308.9 \pm 1.4$ |
| Gemma-3-4B-In | $320.8 \pm 5.6$ |
| Ministral-8B-Instruct-2410 | $337.8 \pm 4.29$ |

### 3.2. Misreporting Optimally Without Raising Suspicion Is Hard

Given a generated output sequence $\mathbf{t}$ with $s = \mathtt{str}(\mathbf{t})$, the provider may raise suspicion if they report $\tilde{\mathbf{t}}_{\max}$, as defined in Eq. 3, because the probability that an LLM actually generates $\tilde{\mathbf{t}}_{\max}$ may be negligible in practice. In fact, if the provider implements procedures to prevent the generation of low-probability tokens, as commonly done in practice, the reported output sequence $\tilde{\mathbf{t}}_{\max}$ may be implausible, as exemplified in Figure 1 for top-$p$ sampling. This lends support to the idea that the provider should not only be required to report an output sequence, but also the next-token probability corresponding to each token in the sequence, offering the user the means to contest a reported output token sequence.

In what follows, we will focus on a setting in which the provider implements top-$p$ sampling (Holtzman et al., 2020), a widely used sampling technique that, given a (partial) token sequence $\mathbf{t}$, restricts the sampling of the next token to a set of tokens to the smallest set $\mathcal{V}_p(\mathbf{t}) \subseteq \mathcal{V}$ whose cumulative next-token probability is at least $p \in (0, 1)$, and aims to find the longest plausible tokenization $\tilde{\mathbf{t}}$ of $s$, *i.e.*,

$$\max_{\tilde{\mathbf{t}} \in \mathcal{V}_s^*} \quad \mathtt{len}\left(\tilde{\mathbf{t}}\right)$$
$$\text{subject to} \quad \tilde{t}_i \in \mathcal{V}_p(\tilde{\mathbf{t}}_{\leq i-1}) \ \forall i \in [\mathtt{len}\left(\tilde{\mathbf{t}}\right)], \quad (4)$$

where $\tilde{\mathbf{t}}_{\leq i-1} = (\tilde{t}_1, \ldots, \tilde{t}_{i-1})$ is the prefix of the reported output sequence up to the $i$-th token.

The following theorem tells us that, in general, the provider cannot expect to solve the problem of finding the longest plausible tokenization under top-$p$ sampling in polynomial time:[6]

**Theorem 3.2.** *The problem of finding the longest tokenization of a given output that is plausible under top-$p$ sampling, as defined in Eq. 4, is NP-Hard.*

---

[6]All proofs of theorems and propositions can be found in Appendix B.

The proof of the above theorem relies on a reduction from the Hamiltonian path problem (Karp, 1972). More specifically, given a graph, it creates an instance of our problem that establishes a one-to-one correspondence between a path that does not visit any node twice and a token sequence that is plausible only if it does not include any token twice. In Appendix B.1.1, we show that the above hardness result can be extended to a setting in which the provider implements top-$k$ sampling and, in Appendix B.1.2, we show that it can also be extended to a setting in which the provider does not implement any procedure to prevent the generation of low-probability tokens but aims to report sequences whose generation probability is greater than a given threshold.

Further, the above hardness result readily implies that there exists a computational barrier that precludes an unfaithful provider from optimally benefiting from misreporting without raising suspicion. However, we will next demonstrate that, in practice, it does not rule out the possibility that a provider efficiently finds and (mis-)reports plausible tokenizations $\tilde{\mathbf{t}}$ longer than $\mathbf{t}$.

### 3.3. Can a Provider Overcharge a User Without Raising Suspicion?

We answer this question affirmatively. As a proof-of-concept, we introduce a simple heuristic algorithm that, given a generated output sequence $\mathbf{t}$ with $s = \mathtt{str}(\mathbf{t})$, efficiently finds a plausible tokenization $\tilde{\mathbf{t}}$ of $s$ longer than or equal to $\mathbf{t}$. Here, our goal is to demonstrate that, under the pay-per-token pricing mechanism predominantly used by cloud providers of LLM-as-a-service, users are indeed vulnerable to self-serving providers who may overcharge them without raising suspicion.

Our heuristic algorithm, summarized in Algorithm 1, is based on the key empirical observation that, given the most likely tokenization $\mathbf{t}$ of a string $s = \mathtt{str}(\mathbf{t})$, alternative tokenizations of $s$ that are not *too different* from $\mathbf{t}$ are very likely to be plausible, as exemplified by Figure 1. In a nutshell, our algorithm starts from a given output sequence $\mathbf{t}$ and iteratively splits tokens in it for a number of iterations $m$ specified by the provider. In each iteration, the algorithm selects the token with the highest index in the vocabulary and, if it is longer than one character, it splits it into a pair of new tokens with the highest minimum index in the vocabulary whose concatenation maps to the same string.[7] The algorithm continues either until it has performed $m$ splits or the selected token is a single character, in which case it terminates the loop. Finally, it checks whether the resulting token sequence $\hat{\mathbf{t}}$ is plausible and, if it is indeed plausible,

---

[7]We focus on splitting tokens based on their index motivated by the BPE algorithm, where tokens with higher indices are (generally) longer, and hence are more likely to result in a plausible tokenization. Refer to Appendix C.2 for concrete examples of how our heuristic modifies token sequences.

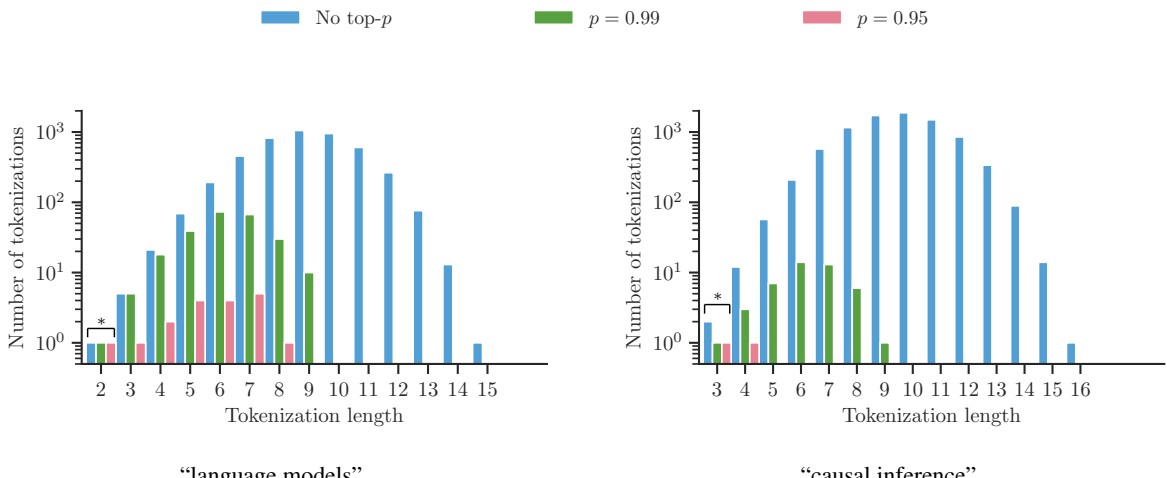

*Figure 1.* **Distribution of tokenizations for two different output strings using the tokenizer of `Llama-3.2-1B-Instruct`.** The panels show the distribution of the length of plausible token sequences for two output strings under top-$p$ sampling for two different values of $p$ and under standard sampling ("No top-$p$"). Here, we set the temperature of the model to 1.0, and denote the most likely tokenization of the string using an asterisk ("*").

it reports it to the user. For example, under top-$p$ sampling, evaluating plausibility reduces to checking whether $\hat{t}_i \in \mathcal{V}_p(\hat{\mathbf{t}}_{\leq i-1})$ for all $i \in [\text{len}(\hat{\mathbf{t}})]$. However, our algorithm is agnostic to the choice of plausibility criteria (refer to Appendices B.1.1 and B.1.2 for alternatives). If $\hat{\mathbf{t}}$ is not plausible, the algorithm reports the true output token sequence $\mathbf{t}$.

Importantly, an efficient implementation of Algorithm 1 has a complexity of $\mathcal{O}(m(\log m + \sigma_{\max}))$, where $\sigma_{\max}$ is the number of characters in the longest token in the vocabulary, and it requires to evaluate the plausibility of a single token sequence—the resulting token sequence $\hat{\mathbf{t}}$. In that context, note that a provider can evaluate the plausibility of a token sequence in a single forward pass of the model, as in speculative sampling (Jie et al., 2025; Vaswani et al., 2023). As a consequence, we argue that, from the provider's perspective, the cost of running Algorithm 1 is negligible in comparison with the monetary reward due to overcharged tokens.

Using prompts from the LMSYS Chatbot Arena platform, we find empirical evidence that, despite its simplicity, Algorithm 1 succeeds at helping a provider overcharge users whenever they serve LLMs with temperature values >1.0, as those commonly used in creative writing tasks. Figure 2 summarizes the results for two LLMs under top-$p$ sampling and temperature 1.3. We find that, for `Llama-3.2-1B-Instruct`, a provider who uses Algorithm 1 can overcharge users by up to 9.5%, 1.6% and 0.3%, and, for `Ministral-8B-Instruct-2410`, they can overcharge by up to 13%, 2.6%, and 0.3%, respectively for $p = 0.99, 0.95, 0.9$. Moreover, we also find that the finan-

**Algorithm 1** It returns a plausible token sequence $\tilde{\mathbf{t}}$ with length greater or equal than the length of $\mathbf{t}$

**Input** True output token sequence $\mathbf{t}$, number of iterations $m$, token-to-id function $\text{id}(\bullet)$

**Initialize** $\hat{\mathbf{t}} \leftarrow \mathbf{t}$

**for** $m$ iterations **do**

  $i \leftarrow \text{argmax}_{j \in [\text{len}(\hat{\mathbf{t}})]} \, \text{id}(\hat{t}_j)$ ▷ Pick the token with the highest index

  **if** $|\text{str}(\hat{t}_i)| = 1$ **then**

    **break** ▷ If it corresponds to a single character, terminate the loop

  **end if**

  $(t'_1, t'_2) \leftarrow \underset{\substack{v_1, v_2 \in \mathcal{V}: \\ \text{str}((v_1, v_2)) = \text{str}(\hat{t}_i)}}{\text{argmax}} \min(\text{id}(v_1), \text{id}(v_2))$

  $\hat{\mathbf{t}} \leftarrow (\hat{\mathbf{t}}_{<i}, t'_1, t'_2, \hat{\mathbf{t}}_{>i})$ ▷ If not, split it into a pair of tokens with the max-min index

**end for**

**if** $\text{plausible}(\hat{t})$ **then**

  $\tilde{\mathbf{t}} \leftarrow \hat{\mathbf{t}}$ ▷ If the resulting token sequence is plausible, report it to the user

**else**

  $\tilde{\mathbf{t}} \leftarrow \mathbf{t}$ ▷ If not, report the true output token sequence

**end if**

**return** $\tilde{\mathbf{t}}$

cial gain is unimodal with respect to the number of iterations $m$ and the optimal value of $m$ decreases as $p$ decreases and achieving plausibility becomes harder. This is because, for large values of $m$, the token sequence $\hat{\mathbf{t}}$ resulting from itera-

tively splitting tokens, becomes less likely to be plausible, as shown in Figure 3 in Appendix C.1. However, if plausible, it does provide a strictly larger financial gain.

The above empirical results demonstrate that there exist efficient and easy-to-implement algorithms that allow a provider to overcharge users without raising suspicion, leaving users vulnerable to the (potentially) malicious behavior of providers. To address this vulnerability, in the next section, we introduce a pricing mechanism that eliminates the provider's incentive to misreport an output token sequence, by design.

## 4. An Incentive-Compatible Pricing Mechanism

To eliminate the provider's incentive to misreport an output token sequence, in this section, we look into the design of incentive-compatible pricing mechanisms. Incentive-compatibility is a (desirable) property studied in mechanism design (Nisan & Ronen, 2001) that, in the context of our work, ensures that the pricing mechanism creates no economic incentive for the provider to misreport an output token sequence—they cannot benefit from not telling the truth.[8]

**Definition 4.1.** A pricing mechanism $r$ is incentive-compatible if and only if, for any generated output token sequence $\mathbf{t} \in \mathcal{V}^*$ and any reported output token sequence $\tilde{\mathbf{t}} \in \mathcal{V}^*$, it holds that $U_{\text{provider}}(\mathbf{t}, \mathbf{t}) \geq U_{\text{provider}}(\tilde{\mathbf{t}}, \mathbf{t})$.

Importantly, if a pricing mechanism satisfies incentive-compatibility, the monetary reward a provider receives for reporting an output token sequence $\tilde{\mathbf{t}}$ depends only on the string $s = \text{str}(\tilde{\mathbf{t}})$ and not on the token sequence itself, as shown by the following proposition:

**Proposition 4.2.** *If a pricing mechanism $r$ is incentive-compatible, then, for all $\hat{\mathbf{t}}, \mathbf{t}' \in \mathcal{V}^*$ such that $\text{str}(\hat{\mathbf{t}}) = \text{str}(\mathbf{t}')$, it holds that $r(\hat{\mathbf{t}}) = r(\mathbf{t}')$.*

Perhaps surprisingly, the above proposition readily allows us to provide a simple characterization of the family of incentive-compatible pricing mechanisms. In particular, the following theorem tells us that it consists of all mechanisms that charge for an output sequence $\mathbf{t}$ linearly on its character counts:

**Theorem 4.3.** *A pricing mechanism $r$ is additive and incentive-compatible if and only if*

$$r(\mathbf{t}) = \sum_{\sigma \in \Sigma} \text{count}_\sigma(\mathbf{t}) \cdot r(\sigma) \text{ for all } \mathbf{t} \in \mathcal{V}, \quad (5)$$

*where $\text{count}_\sigma(\mathbf{t})$ counts the number of occurrences of the character $\sigma$ in $\text{str}(\mathbf{t})$.*

---

[8]In the mechanism design literature, an incentive-compatible mechanism is also called truthful or strategy-proof.

As an immediate consequence, if the provider decides to assign the same price $r_c$ to each character $\sigma \in \Sigma$, there exists only one incentive-compatible pricing mechanism, *i.e.*, $r(\mathbf{t}) = |\text{str}(\mathbf{t})| \cdot r_c$, which we refer to as the pay-per-character pricing mechanism.

**Implementation and downstream effects of pay-per-character.** The pay-per-character pricing mechanism is a simple solution to the problem of misreporting output token sequences. However, in practice, both providers and users may like to avoid financial overheads from transitioning from the pay-per-token to the pay-per-character pricing mechanism. In this context, one simple way to reduce the overheads is to set the price of a single character to $r_c = r_0/\texttt{cpt}$, where $r_0$ is the price of a single token under the provider's current pay-per-token pricing mechanism and $\texttt{cpt}$ is the (empirical) average number of characters per token across the responses to user prompts. For instance, in the responses to prompts from the LMSYS Chatbot Arena platform used in our experiments, the average number of characters per token is $\texttt{cpt} = 4.50$ for LLMs in the `Llama` family, $\texttt{cpt} = 4.22$ for the `Gemma` family and $\texttt{cpt} = 4.43$ for the `Ministral` family. This would ensure that, in expectation, the provider's revenue and the users' cost are the same under both pricing mechanisms.

Moreover, transitioning from a pay-per-token to the pay-per-character pricing mechanism creates positive incentives for providers that choose to truthfully report the generated token sequence. Indeed, under pay-per-token, given two token sequences $\mathbf{t}$ and $\mathbf{t}'$ such that $\text{str}(\mathbf{t}) = \text{str}(\mathbf{t}')$, a provider that faithfully reports tokenizations would have higher utility when the longest sequence amongst $\mathbf{t}$ and $\mathbf{t}'$ is generated. On the contrary, for a faithful provider under the pay-per-character pricing mechanism, it holds that $U_{\text{provider}}(\mathbf{t}, \mathbf{t}) > U_{\text{provider}}(\mathbf{t}', \mathbf{t}')$ whenever $\text{len}(\mathbf{t}) < \text{len}(\mathbf{t}')$. In other words, a provider that never misreports has a clear incentive to generate the shortest possible output token sequence, and improve current tokenization algorithms such as BPE, so that they compress the output token sequence as much as possible (Petrov et al., 2023). Such improvements would not only benefit the provider by increasing their utility but also have significant positive downstream effects, such as reduced energy consumption, faster inference, and better use of limited context windows.

## 5. Discussion and Limitations

In this section, we highlight several limitations of our work, discuss its broader impact, and propose avenues for future work.

**Model assumptions.** We have focused on additive pricing mechanisms, which include the widely used pay-per-token mechanism. It would be interesting to analyze provider

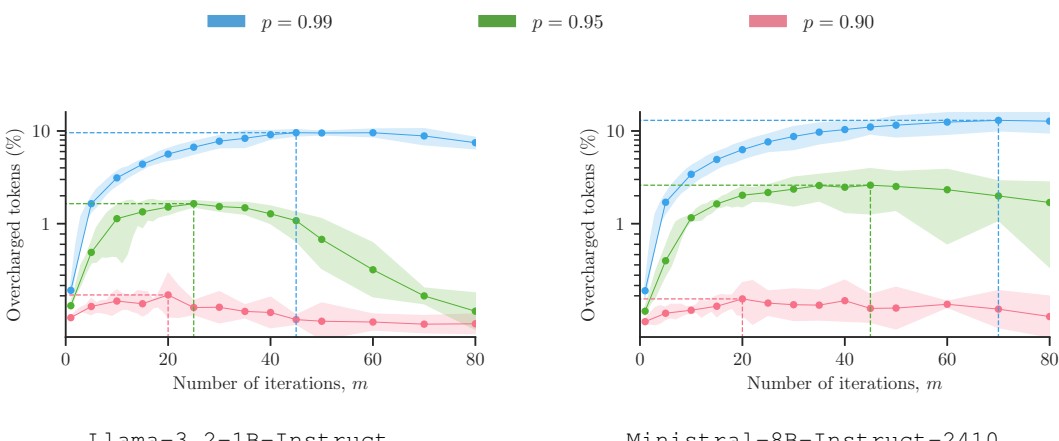

*Figure 2.* **Financial gain from misreporting the tokenization of outputs using Algorithm 1.** The panels show the percentage of tokens overcharged by an unfaithful provider who misreports the tokenization of the outputs generated by an LLM to 400 prompts from the LMSYS Chatbot Arena platform using Algorithm 1, for different values of $m$ and $p$. Here, we set the temperature of the model to 1.3 and repeat each experiment 5 times to obtain 90% confidence intervals. Refer to Appendix C.1 for additional results using alternative temperature values and other LLMs.

incentives under other families of pricing mechanisms proposed in the literature, such as those based on the quality of the generated text (Saig et al., 2024). In this context, a natural direction is to design a pricing mechanism that simultaneously incentivizes multiple desirable behaviors, such as faithful token reporting and output quality. Moreover, we have assumed that the provider pays a negligible cost for evaluating the plausibility of a token sequence, as Algorithm 1 only performs such an evaluation once. However, the design of more complex algorithms performing multiple evaluations should consider the trade-off between the additional profit obtained by using the algorithm against the cost of running it. Further, in the context of contract theory, a principal typically designs a contract in order to disincentivize the agent from taking hidden unwanted actions (Dütting et al., 2024). In our case, the provider (*i.e.*, the agent) is the one who both designs the pricing mechanism (*i.e.*, the contract) and has the power to take hidden actions, leaving the user with limited leverage. Practically, a shift from pay-per-token to other pricing mechanisms, such as pay-per-character, would require external regulation (or user pressure). In this context, we note that while the hardness result in Theorem 3.2 holds without additional assumptions, providers would arguably only be forced to report plausible tokenizations if the users have a mechanism to verify that a token sequence is in fact plausible. This raises the question of rigorously defining the information that both parties have access to when interacting, and how to verify that tokens are generated according to the advertised inference process. We leave this for future work and point towards Trusted Execution Environments (Li et al., 2023)

and Zero-Knowledge Proofs (Sun et al., 2024) as potential solutions to this problem.

**Methods.** To demonstrate the vulnerability of users under the pay-per-token pricing mechanism, we have introduced a heuristic algorithm that allows the provider to increase their profit by finding longer yet plausible tokenizations of the true output token sequence. However, there may exist other, more sophisticated methods for the provider to take advantage of the pay-per-token pricing mechanism, and there may also exist ways to defend users against such malicious behavior, other than a change of the pricing mechanism. Further, misreporting the tokenization of an output sequence is not the only type of strategic behavior that the provider can exhibit, as they have the capacity to misreport other elements of the generative process, such as the next-token distributions or the output string. It would be interesting to explore the implications of these other types of attacks, as well as the potential for auditing them, for example, by detecting whether there is a mismatch between the next-token distributions and the frequencies of the tokens over multiple generations. Lastly, we note that even if pay-per-character provably disincentivizes token misreporting, it does not prevent the provider from artificially increasing the number of characters in the output string. However, arguably, this would affect the quality of the generated text, which would be reflected in the user experience and could lead to a loss of reputation for the provider.

**Evaluation.** We have conducted experiments with state-of-the-art open-weights LLMs from the Llama, Gemma and Ministral families, using different tokenizers and archi-

tectures. It would be interesting to evaluate the possibility of misreporting in proprietary LLMs, which are widely used in practice. Further, we have illustrated our theoretical results using prompts from the LMSYS Chatbot Arena platform. Although this platform is arguably the most widely used for LLM evaluation based on pairwise comparisons, it is important to note that it has been recently criticized (Singh et al., 2025; Zhou et al., 2023), and the prompts submitted to it may not be representative of the real-world distribution of user prompts. Lastly, it would be interesting to analyze in detail the potential implications of transitioning from pay-per-token to pay-per-character pricing. As a proof of concept, we have shown how character pricing can be easily derived from token pricing; this conversion may depend on model specifications and query distribution.

## 6. Conclusions

In this work, we have studied the financial incentives of cloud-based providers in LLM-as-a-service using a principal-agent model of delegated autoregressive generation. We have demonstrated that the widely used pay-per-token pricing mechanism incentivizes a provider to misreport the tokenization of the outputs generated by the LLM they serve. We have shown that, if the provider is required to be transparent about the generative process used by the LLM, it is provably hard for the provider to optimally benefit from misreporting without raising suspicion. However, we have introduced an efficient algorithm that, in practice, allows a transparent provider to benefit from misreporting, overcharging users significantly without raising suspicion. To address this vulnerability, we have introduced a simple incentive-compatible pricing mechanism, pay-per-character, which eliminates the financial incentive for misreporting tokenizations. We hope that our work will raise awareness that, under pay-per-token, users of LLM-as-a-service are vulnerable to (unfaithful) providers, and encourage a paradigm shift towards alternative pricing mechanisms, such as pay-per-character.

## Impact statement

Our work sheds light on the perverse incentives that arise from the pay-per-token pricing mechanism, which is the most widely used pricing mechanism in the context of LLM-as-a-service. On the positive side, we believe that our work can spark a discussion on the need for more transparent and fair pricing mechanisms in the LLM ecosystem. On the flip side, the heuristic algorithm we introduce could be misused by a malicious provider to overcharge users. However, we emphasize that our intention is to use it as a proof-of-concept, and not as an algorithm to be deployed in practice, similarly to the broader literature on adversarial attacks in machine learning (Szegedy et al., 2013; Goodfellow et al., 2014; Chakraborty et al., 2021).

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

# A. Additional Experimental Details

Here, we provide additional details on the experimental setup, including the hardware used, the dataset and models used, as well as details on the generation process.

**Hardware setup.** Our experiments are executed on a compute server equipped with $2 \times$ Intel Xeon Gold 5317 CPU, 1,024 GB main memory, and $2 \times$ A100 Nvidia Tesla GPU (80 GB, Ampere Architecture). In each experiment, a single Nvidia A100 GPU is used.

**Datasets.** For the results presented in Figure 2, Table 1 and Appendix C.1 we generated model responses to prompts obtained from the LMSYS-Chat-1M dataset (Zheng et al., 2024). We use the LMSYS-Chat-1M dataset exclusively to obtain a varied sample of potential user prompts. We filter user prompts to obtain the 400 first questions that are in English language (by using the `language` keyword) and whose length (in number of characters) is in the range $[20, 100]$, to avoid trivial or overly elaborated prompts.

**Models.** In our experiments, we use the models `Llama-3.2-3B-Instruct` and `Llama-3.2-3B-Instruct` from the `Llama` family, the models `Gemma-3-1B-It` and `Gemma-3-4B-It` from the `Gemma` family, and `Ministral-8BInstruct-2410`. The models are obtained from publicly available repositories from `Hugging Face`[9].

**Generation details.** For the experiments in Figure 1, we run an exhaustive search over all possible tokenizations for each string, reporting the distribution of their length under the name "No top-$p$". For every tokenization, we make a forward pass with the model `Llama-3.2-1B-Instruct` to obtain the token probabilities from the combination of prompt and token sequence. We then verify if the token sequence is plausible under top-$p$ sampling with temperature 1 and various values of the parameter $p$. Note that since this is a deterministic process, we do not report any error bars.

For the experiments involving the LMSYS dataset, we use the `transformers` library in `Python 3.11` to generate outputs of varying length between 200 and 300 tokens under various temperature and $p$ values. Each model generates a total of 2000 output token sequences for the first 400 filtered prompts of the LMSYS dataset, by running 5 independent generations with different seeds. We then compute standard deviations across the 5 repetitions, and $90\%$ symmetric confidence intervals for the mean values assuming a $t-$distribution value of 2.015. The $90\%$ confidence intervals are shown in the plots and table.

**Licenses.** The LMSYS-Chat-1M dataset is licensed under the LMSYS-Chat-1M Dataset License Agreement.[10] The `Llama-3.2-1B-Instruct` and `Llama-3.2-3B-Instruct` models are licensed under the LLAMA 3.2 COMMUNITY LICENSE AGREEMENT.[11]. The `Gemma-3-1B-It` and `Gemma-3-4B-It` models are licensed under the GEMMA TERMS OF USE.[12]. The `Ministral-8B-Instruct-2410` model is licensed under the MISTRAL AI RESEARCH LICENSE.[13].

---

[9]https://huggingface.co/meta-llama/Llama-3.2-3B-Instruct
https://huggingface.co/meta-llama/Llama-3.2-1B-Instruct
https://huggingface.co/google/gemma-3-1b-it
https://huggingface.co/google/gemma-3-4b-it
https://huggingface.co/mistralai/Ministral-8B-Instruct-2410
[10]https://huggingface.co/datasets/lmsys/lmsys-chat-1m
[11]https://ai.google.dev/gemma/terms
[12]https://www.gemma.com/gemma3_0/license/
[13]https://mistral.ai/static/licenses/MRL-0.1.md

# B. Proofs

## B.1. Proof of Theorem 3.2

We prove the theorem by reduction from the Hamiltonian path problem (Karp, 2010), which is known to be NP-complete, to the problem of finding a plausible tokenization under top-$p$ sampling longer than a given number of tokens. Consequently, this will prove the hardness of the problem of finding a longest plausible token sequence $\tilde{\mathbf{t}}$ under top-$p$ sampling, as stated in Eq. 4. In the Hamiltonian path problem, we are given a directed graph $\mathcal{G}$, that is, a set of nodes $\mathcal{N} = \{1, \ldots, n\}$ and a set of edges $\mathcal{E}$ between them, where $e = (\nu, \nu')$ denotes an edge from node $\nu$ to node $\nu'$. The goal is to decide whether there exists a path that visits all nodes exactly once.

The core idea of the construction is to represent a path in the graph $\mathcal{G}$ as a sequence of tokens, where each node $j \in \mathcal{N}$ is represented by a token consisting of $j$ times the character "a". In addition, we set the parameter $p \in (0, 1)$ of top-$p$ sampling and the next-token distributions of the LLM such that a token sequence $\tilde{\mathbf{t}}$ with $\mathrm{str}\left(\tilde{\mathbf{t}}\right) = \mathrm{str}(\mathbf{t})$ and $\mathrm{len}\left(\tilde{\mathbf{t}}\right) > 1$ is plausible if and only if the tokens in $\tilde{\mathbf{t}}$ correspond to a Hamiltonian path in the graph $\mathcal{G}$.

We proceed with the construction as follows. Let $\Sigma = \{\text{"a"}\}$ be the alphabet and the LLM's vocabulary be

$$\mathcal{V} = \{\text{"a"}, \text{"aa"}, \ldots, \underbrace{\text{"a...a"}}_{n \text{ times}}, \underbrace{\text{"a...a"}}_{\lambda \text{ times}}, \varnothing\},$$

where $\lambda = \sum_{j=1}^{n} j = n(n+1)/2$ and $\varnothing$ denotes the end-of-sequence token. Moreover, let the true output token sequence $\mathbf{t}$ consist of a single token—the one that contains $\lambda$ times the character "a". Further, to keep the notation concise, we refer to the set of the first $n$ tokens in $\mathcal{V}$ as $\mathcal{V}_n$. Then, we define a mapping $\Phi \colon \mathcal{V}_n \to \mathcal{N}$ from tokens to nodes as

$$\Phi(\underbrace{\text{"a...a"}}_{j \text{ times}}) = j \text{ for } j = 1, \ldots, n.$$

We fix the parameter $p$ and a next-token distribution of the LLM such that, given a (partial) token sequence $\tilde{\mathbf{t}} = \left(\tilde{t}_1, \ldots, \tilde{t}_k\right)$, the restricted set of tokens $\mathcal{V}_p\left(\tilde{\mathbf{t}}\right)$ from which the LLM can sample the next token is given by

$$\mathcal{V}_p\left(\tilde{\mathbf{t}}\right) = \begin{cases} \{\varnothing\} & \text{if } \left|\mathrm{str}\left(\tilde{\mathbf{t}}\right)\right| \geq \lambda \\ \mathcal{V} \setminus \varnothing & \text{if } \tilde{\mathbf{t}} = () \\ \left\{v \in \mathcal{V}_n : v \neq \tilde{t}_i \text{ for all } i \in [k] \text{ and } \left(\Phi\left(\tilde{t}_k\right), \Phi\left(v\right)\right) \in \mathcal{E}\right\} \cup \{\varnothing\} & \text{otherwise.} \end{cases} \quad (6)$$

In words, the last case states that the LLM can sample any token consisting of up to $n$ times the character "a" as long as it is not already in the sequence $\tilde{\mathbf{t}}$, that is, the corresponding node has not been visited yet, and there is an edge in the graph $\mathcal{G}$ connecting that node to the node corresponding to the last token in $\tilde{\mathbf{t}}$. When the sequence $\tilde{\mathbf{t}}$ is empty (*i.e.*, the path has not started yet), the LLM can sample any token in $\mathcal{V}$ except for the end-of-sequence token $\varnothing$, which it is only allowed to sample when the sequence $\tilde{\mathbf{t}}$ contains at least $\lambda$ characters.

We can now show that a Hamiltonian path in the graph $\mathcal{G}$ exists if and only if the solution $\tilde{\mathbf{t}}$ to the optimization problem given by Eq. 4 has $\mathrm{len}\left(\tilde{\mathbf{t}}\right) > 1$.[14] Assume that the optimal solution to the problem is such that $\mathrm{len}\left(\tilde{\mathbf{t}}\right) > 1$. Then, $\tilde{\mathbf{t}}$ cannot contain the token that consists of $\lambda$ times the character "a" because this would imply that it consists of strictly more than $\lambda$ characters and, therefore, $\mathrm{str}(\tilde{\mathbf{t}}) \neq \mathrm{str}(\mathbf{t})$. Additionally, $\tilde{\mathbf{t}}$ cannot contain any token twice as that would violate its plausibility according to Eq. 6. Therefore, it has to hold that $\tilde{\mathbf{t}}$ contains all tokens in $\mathcal{V}_n$ exactly once, since this is the only way to form a sequence that contains $\lambda = \sum_{j=1}^{n} j$ characters. This implies that there exists a sequence of edges $\left(\Phi\left(\tilde{t}_1\right), \Phi\left(\tilde{t}_2\right)\right), \ldots, \left(\Phi\left(\tilde{t}_{n-1}\right), \Phi\left(\tilde{t}_n\right)\right)$ in the graph $\mathcal{G}$ that visits all nodes exactly once. Hence, a Hamiltonian path exists.

Now, assume that there exists a Hamiltonian path in the graph $\mathcal{G}$ that visits all nodes once, forming a sequence $(\nu_1, \nu_2, \ldots, \nu_n)$ with $\nu_i \in \mathcal{N}$ and $\nu_i \neq \nu_j$ for $i \neq j$. Then, the corresponding token sequence $\mathbf{t}' = (t'_1, t'_2, \ldots, t'_n)$ with $\Phi\left(t'_i\right) = \nu_i$ for $i \in [n]$ is a valid tokenization of the string $\mathrm{str}(\mathbf{t})$ since $\sum_{i=1}^{n} |\mathrm{str}(t'_i)| = \sum_{i=1}^{n} \nu_i = \lambda$. Moreover, the sequence $\mathbf{t}'$ is plausible by construction and satisfies $\mathrm{len}\left(\mathbf{t}'\right) = n > 1 = \mathrm{len}\left(\tilde{\mathbf{t}}\right)$. Finally, note that if $\mathcal{G}$ does not admit a Hamiltonian path, then $\mathrm{str}(\mathbf{t})$ cannot be tokenized as a sequence of plausible tokens in $\mathcal{V}_n$. Hence, the only plausible tokenization is the token with $\lambda$ characters, which has length 1. This concludes the proof.

---

[14]For ease of exposition, we assume that the end-of-sequence token $\varnothing$ does not contribute to the length of the sequence $\tilde{\mathbf{t}}$.

In what follows, we present two extensions of the reduction to other settings where a provider may want to misreport the output token sequence without raising suspicion. Specifically, we consider the case where the provider reports a token sequence $\tilde{\mathbf{t}}$ that is plausible under top-$k$ sampling and the case where the provider reports a token sequence $\tilde{\mathbf{t}}$ whose probability is greater than a given threshold.

### B.1.1. HARDNESS OF FINDING THE LONGEST PLAUSIBLE TOKENIZATION UNDER TOP-$k$ SAMPLING

Top-$k$ sampling is an approach of filtering out low-probability tokens during the sampling process, similar to top-$p$ sampling. In top-$k$ sampling, given a partial token sequence $\tilde{\mathbf{t}}$, the LLM samples the next token from the set of $k$ most probable tokens $\mathcal{V}_k\left(\tilde{\mathbf{t}}\right)$ at each step of the autoregressive process, where $k \in \{1, \ldots, |\mathcal{V}| - 1\}$. In this setting, the problem of finding a longest tokenization of a given output token sequence $\mathbf{t}$ that is plausible under top-$k$ sampling is NP-Hard with the core idea of the reduction being similar to the one for top-$p$ sampling.

The main difference lies in the fact that, in top-$k$ sampling, the restricted set of tokens $\mathcal{V}_k\left(\tilde{\mathbf{t}}\right)$ needs to have a fixed size $k$ in contrast to the construction of $\mathcal{V}_p\left(\tilde{\mathbf{t}}\right)$ in Eq. 6, which is a variable size set. To ensure that similar arguments for establishing a one-to-one correspondence between a Hamiltonian path in the graph $\mathcal{G}$ and a plausible token sequence $\tilde{\mathbf{t}}$ of length greater than 1 still hold, one can construct the set $\mathcal{V}_k\left(\tilde{\mathbf{t}}\right)$ using a similar approach as in Eq. 6 but also including "padding" tokens that do not correspond to any node in the graph $\mathcal{G}$ to maintain a fixed size. To this end, we can maintain the same true output token sequence $\mathbf{t}$, consisting of $n(n+1)/2$ times "a" and augment the vocabulary $\mathcal{V}$ of the previous construction by adding $n$ additional tokens

$$\mathcal{V}_b = \{\text{``b''}, \text{``bb''}, \ldots, \underbrace{\text{``b...b''}}_{n \text{ times}}\}$$

that are irrelevant for the string $s = \text{str}(\mathbf{t})$, do not correspond to any node in the graph $\mathcal{G}$, and do not affect the mapping $\Phi$.

Then, note that, the set $\mathcal{V}_p\left(\tilde{\mathbf{t}}\right)$ in Eq. 6 contains at most $n+1$ tokens. Here, the idea is to set $k = n+1$ and to construct the set $\mathcal{V}_k\left(\tilde{\mathbf{t}}\right)$ as follows:

$$\mathcal{V}_k\left(\tilde{\mathbf{t}}\right) = \mathcal{V}_p\left(\tilde{\mathbf{t}}\right) \cup G\left(\mathcal{V}_p\left(\tilde{\mathbf{t}}\right)\right), \tag{7}$$

where $G\left(\mathcal{V}_p\left(\tilde{\mathbf{t}}\right)\right)$ is the set of the first $n+1 - |\mathcal{V}_p\left(\tilde{\mathbf{t}}\right)|$ tokens in $\mathcal{V}_b$. Since the additional tokens in $G\left(\mathcal{V}_p\left(\tilde{\mathbf{t}}\right)\right)$ are not part of the mapping $\Phi$ and cannot be used to tokenize the string $s = \text{str}(\mathbf{t})$, they influence neither the plausibility of the optimal solution to the problem of Eq. 4 nor the corresponding Hamiltonian path in the graph $\mathcal{G}$. Therefore, the same arguments as in the proof of Theorem 3.2 hold, and we conclude that the problem of finding a longest tokenization of a given output token sequence $\mathbf{t}$ that is plausible under top-$k$ sampling is NP-Hard.

### B.1.2. HARDNESS OF FINDING THE LONGEST TOKENIZATION WHOSE GENERATION PROBABILITY IS GREATER THAN A THRESHOLD

We now focus on a slightly different setting where the provider reports a token sequence $\tilde{\mathbf{t}}$ under the plausibility condition that the LLM does not assign very low probability to the sequence as a whole. Formally, we require that the probability of the LLM generating the token sequence $\tilde{\mathbf{t}}$ satisfies

$$P\left(\tilde{\mathbf{t}}\right) := P\left(\tilde{t}_1\right) \prod_{i=2}^{k} P\left(\tilde{t}_i \mid \tilde{\mathbf{t}}_{<i}\right) \geq \varepsilon, \tag{8}$$

where $\varepsilon$ is a user-specified threshold and $P\left(\tilde{t}_i \mid \tilde{\mathbf{t}}_{<i}\right)$ is the probability of the LLM generating the token $\tilde{t}_i$ given the previously generated tokens $\tilde{\mathbf{t}}_{<i} = \left(\tilde{t}_1, \ldots, \tilde{t}_{i-1}\right)$. In this setting, the problem of finding a longest tokenization under Eq. 8 is also NP-hard. Similarly, as before, the proof is to set the next-token distributions of the LLM in a way that assigns low probability to token sequences that do not lead to a Hamiltonian path in $\mathcal{G}$. Specifically, let $\delta$ be a constant such that $0 < \delta < 1/(n+1)$, and assume all next-token distributions are such that, given $\left(\tilde{t}_1, \ldots, \tilde{t}_k\right)$, assign probability mass $(1-\delta)/n$ to each of the tokens in

$$\mathcal{H}_i := \left\{v \in \mathcal{V}_n : v \neq \tilde{t}_i \text{ for all } i \in [k] \text{ and } \left(\Phi\left(\tilde{t}_k\right), \Phi\left(v\right)\right) \in \mathcal{E}\right\}, \tag{9}$$

$\delta$ to each of the tokens in $\mathcal{V}_n \setminus \mathcal{H}_i$, 0 to the token with $\lambda$ times the character "a", and any remaining probability mass to the end-of-sequence token $\varnothing$.[15] The high-level idea here is to set the probabilities of next tokens in such a way that the LLM

---

[15]Using the assumption that $\delta < 1/(n+1)$, it is easy to verify that the above construction leads to a valid probability distribution.

assigns very low probability to the entire token sequence $\tilde{\mathbf{t}}$ if it concatenates two tokens whose corresponding nodes are not connected via an edge in the graph $\mathcal{G}$ or if the latter token has already been used in the sequence.

Given this construction, we set the user-specified threshold as $\varepsilon = \left(\frac{1-\delta}{n}\right)^n$. Now, given a Hamiltonian path in the graph $\mathcal{G}$ that visits all nodes once and forms a sequence $(\nu_1, \nu_2, \ldots, \nu_n)$ with $\nu_i \in \mathcal{N}$ and $\nu_i \neq \nu_j$ for $i \neq j$, the corresponding token sequence $\mathbf{t}' = (t'_1, t'_2, \ldots, t'_n)$ has cumulative probability exactly $\varepsilon$, so it is plausible and has length greater than 1. Reciprocally, given a plausible tokenization $\tilde{\mathbf{t}}$ with length greater than 1, the corresponding sequence $\left(\Phi\left(\tilde{t}_1\right), \Phi\left(\tilde{t}_2\right)\right), \ldots, \left(\Phi\left(\tilde{t}_{n-1}\right), \Phi\left(\tilde{t}_n\right)\right)$ has to be a Hamiltonian path. If this is not true, at least one of the tokens in $\tilde{\mathbf{t}}$ does not belong in its respective set $\mathcal{H}_i$ defined by Eq. 9, and hence the probability of the sequence $\tilde{\mathbf{t}}$ is at most

$$P\left(\tilde{\mathbf{t}}\right) \leq \delta\left(\frac{1-\delta}{n}\right)^{n-1} < \varepsilon, \tag{10}$$

which contradicts the assumption that $\tilde{\mathbf{t}}$ is plausible.

### B.2. Proof of Proposition 4.2

Let $\mathbf{t} = \hat{\mathbf{t}}$ be the true output sequence generated by the LLM. Then, by Definition 4.1, it holds that

$$U_{\text{provider}}(\hat{\mathbf{t}}, \hat{\mathbf{t}}) \geq U_{\text{provider}}(\mathbf{t}', \hat{\mathbf{t}}) \overset{(*)}{\Longrightarrow} r\left(\hat{\mathbf{t}}\right) - c\left(\hat{\mathbf{t}}\right) \geq r\left(\mathbf{t}'\right) - c\left(\hat{\mathbf{t}}\right) \implies r\left(\hat{\mathbf{t}}\right) \geq r\left(\mathbf{t}'\right),$$

where $(*)$ follows from Eq. 1.

Now, consider that the true output sequence generated by the LLM is $\mathbf{t} = \mathbf{t}'$. Similarly, as before, we have $U(\mathbf{t}', \mathbf{t}') \geq U(\hat{\mathbf{t}}, \mathbf{t}')$, which implies that $r\left(\mathbf{t}'\right) \geq r\left(\hat{\mathbf{t}}\right)$. Combining the two inequalities, we get $r\left(\hat{\mathbf{t}}\right) = r\left(\mathbf{t}'\right)$.

### B.3. Proof of Theorem 4.3

Let $\mathbf{t}' = (t'_1, \ldots, t'_k)$ be the tokenization of the string $s = \text{str}(\mathbf{t})$ that consists only of single-character tokens, *i.e.*, $\text{str}(\mathbf{t}) = \text{str}(\mathbf{t}')$ with $|\text{str}(\mathbf{t}')| = |\text{str}(\mathbf{t})| = k$. Note that such a tokenization exists, since $\Sigma \subseteq \mathcal{V}$. From Proposition 4.2, we get

$$r\left(\mathbf{t}\right) = r\left(\mathbf{t}'\right) \overset{(*)}{=} \sum_{i=1}^{k} r\left(t'_i\right) = \sum_{i=1}^{k} \sum_{\sigma \in \Sigma} \mathbb{1}[t'_i = \sigma] \cdot r(\sigma)$$

$$= \sum_{\sigma \in \Sigma} \text{count}_\sigma\left(\mathbf{t}'\right) \cdot r(\sigma) \overset{(**)}{=} \sum_{\sigma \in \Sigma} \text{count}_\sigma\left(\mathbf{t}\right) \cdot r(\sigma),$$

where $\mathbb{1}$ denotes the indicator function, $(*)$ holds because the pricing mechanism is additive and $(**)$ holds because $\text{str}(\mathbf{t}') = \text{str}(\mathbf{t})$.

# C. Additional Experimental Results

## C.1. Performance of Algorithm 1 under Different LLMs and Temperature Values

In this section, we evaluate Algorithm 1 on outputs generated by five LLMs to the same prompts used in Section 3 under different temperature values.

Figure 3 shows the fraction of generated outputs for which Algorithm 1 finds a longer plausible tokenization. We observe that, the higher the values of $p$ and temperature, the higher the likelihood that Algorithm 1 finds plausible longer tokenizations. Moreover, we also observe that, for outputs given by the Gemma-3-4B-It model, Algorithm 1 is less likely to find plausible longer tokenizations across all temperature and $p$ values. We hypothesize that this is due to the fact that Gemma-3-4B-It is the only model in our experiments that is multimodal and the level of randomness in its next-token distributions may be lower than in the other models.

Figure 4 shows the percentage of tokens overcharged by an unfaithful provider who uses Algorithm1. We observe that the percentage of overcharged tokens is unimodal with respect to the number of iterations $m$, and the higher the value the temperature and $p$, the higher the percentage of overcharged tokens, as the top-$p$ sets become larger and the likelihood that a longer tokenization is plausible increases.

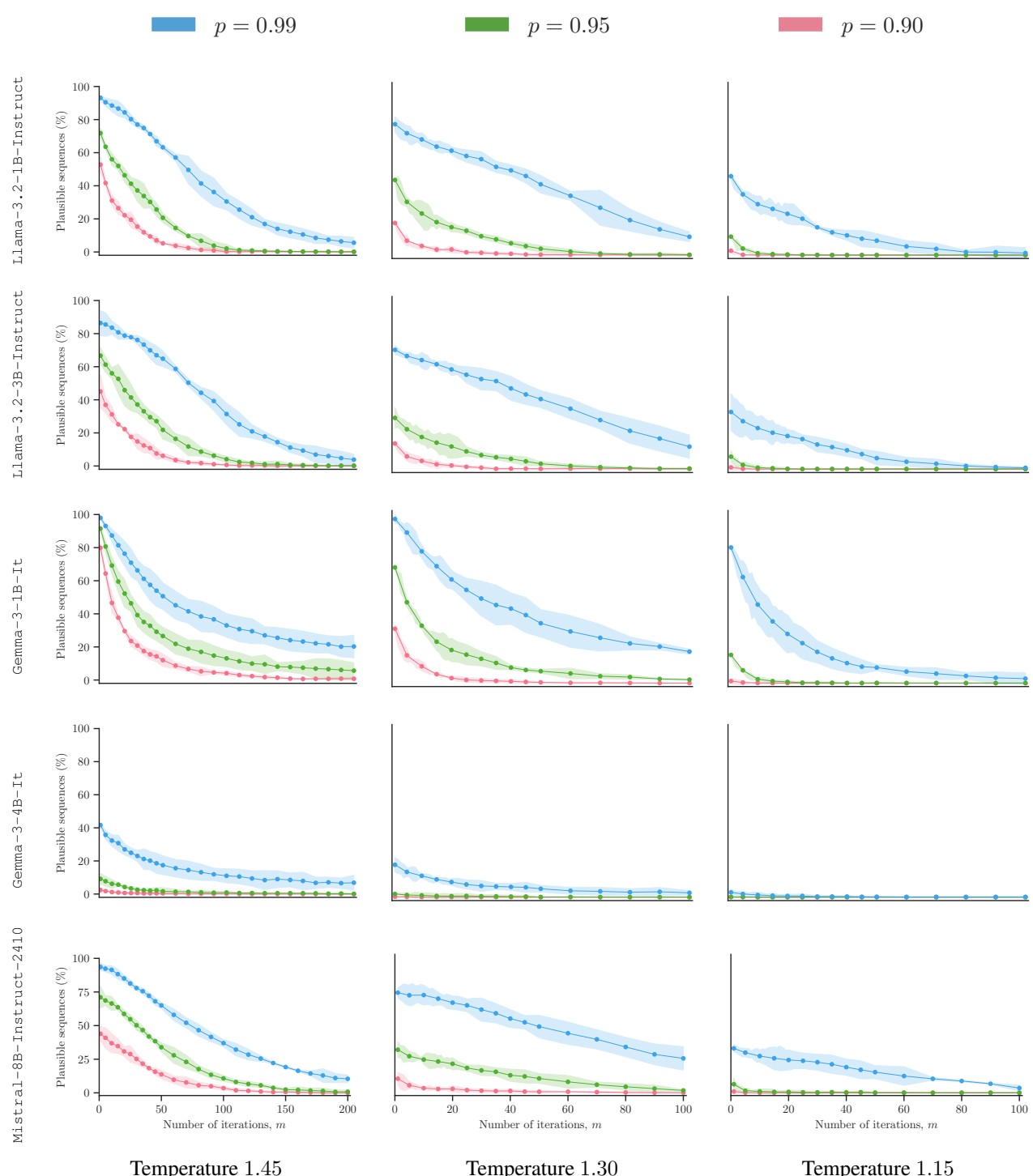

*Figure 3.* **Fraction of generated outputs for which Algorithm 1 finds a plausible longer tokenization.** The figure shows, for different model families, the fraction of token sequences where the heuristic implemented in Algorithm 1 finds a plausible longer tokenization under top$-p$ sampling and various temperature levels, as a function of the additional tokens overcharged to the user (*i.e.*, the number of iterations $m$ in Algorithm 1). The output token sequences $\mathbf{t}$ are generated for the first 400 prompts in the LMSYS dataset. We repeat each experiment 5 times to calculate 90% confidence intervals.

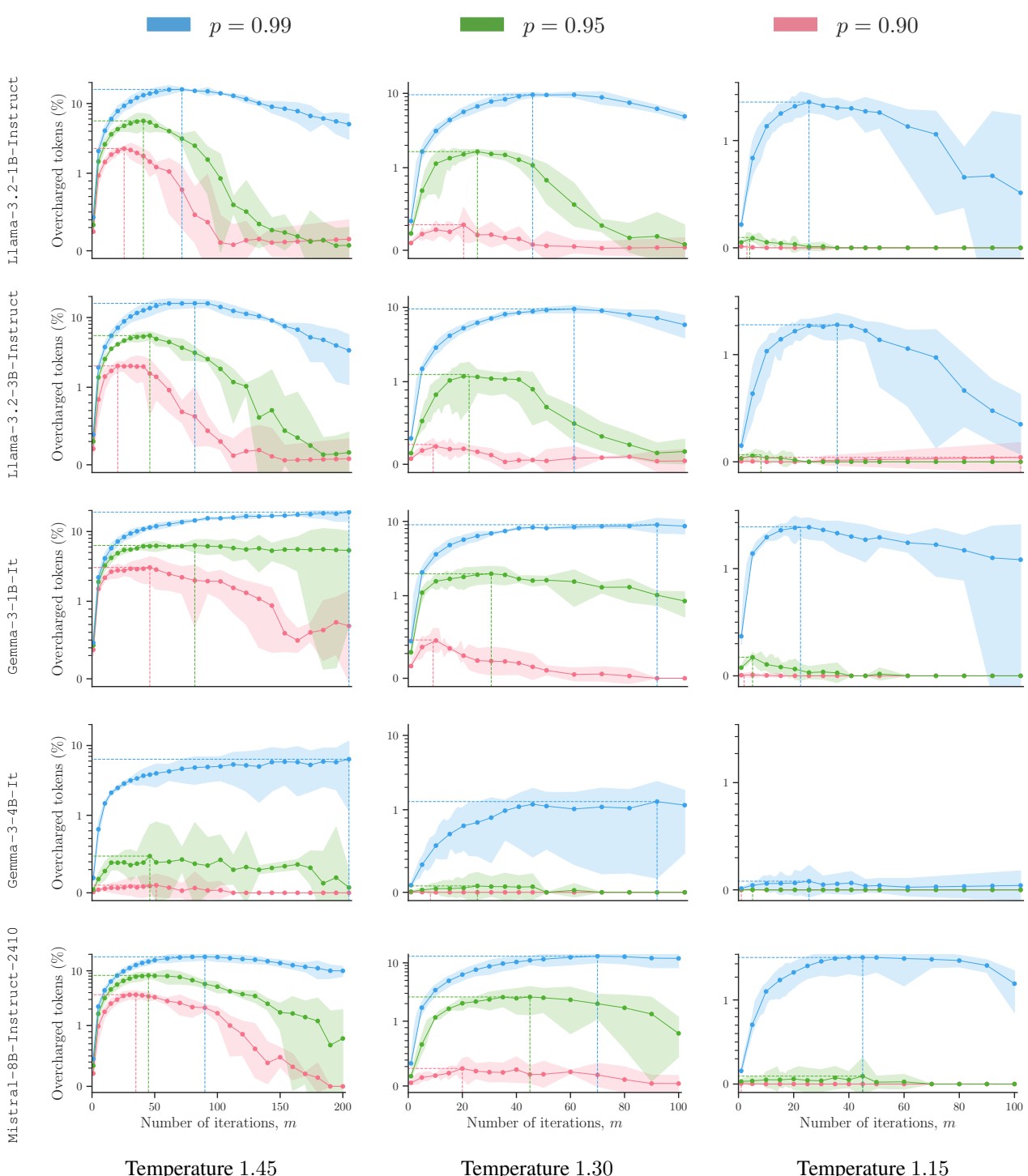

*Figure 4.* **Financial gain from misreporting the tokenization of outputs using Algorithm 1.** The figure shows, across different model families and for the first 400 LMSYS prompts, the total percentage of tokens that a provider using top−$p$ sampling following the heuristic in Algorithm 1 could overcharge the user, as a function of the number of iterations and for various temperature values. Dashed lines correspond to the maximum of each curve. We repeat each experiment 5 times to calculate 90% confidence intervals.

### C.2. Examples of Plausible Output Token Sequences Found by Algorithm 1

To illustrate how Algorithm 1 works, here, we provide examples of output token sequences generated by the `Llama-3.2-1B-Instruct` model, where the algorithm has found plausible tokenizations that are longer than the original output token sequence. Across all examples, we use the `Llama-3.2-1B-Instruct` model and set $p = 0.95$ and the temperature of the model to $1.3$. We select prompts from the LMSYS dataset. For each example, we show (i) the true output token sequence generated by the model, and (ii) the modified output token sequence returned by Algorithm 1. We use "|" to indicate separations between tokens as generated by the model, and we use "|" to indicate the split points of the tokens that result from Algorithm 1. The number above each red separator indicates the iteration of the algorithm in which the respective token was split. We show all iterations until the sequence first becomes non-plausible.

```
...  The| third| film| appears| to| delve| into| the| themes| of| societal|
reaction| and|...  Here| are| movies| that| offer| similar| thematic con-
cerns|...
```

(a) True output token sequence

```
                                                        (1)
...  The| third| film| appears| to| del |ve| into| the| themes| of|
          (2)
soci |etal| reaction| and|...  Here| are| movies| that| offer| similar| the-
                    (3)
matic conce | rns|...
```

(b) Modified output token sequence

*Figure 5.* Responses to the prompt "`is Dead Snow worth watching or should I watch directly Dead Snow 2?`".

```
...  Here| are| a| few| options| :|
1|.| **| T|rello|**:| T|rello| is| a| visual| project| management| tool|...
2|.| **| J|IRA|**:| As| mentioned|,| J|IRA| is| a| popular| At|lassian| suite|...
```

(a) True output token sequence

```
...  Here| are| a| few| options| :|
                    (1)
1|.| **| T|rello|** | :| T|rello| is| a| visual| project| management| tool|...
                  (2)                                              (3)
2|.| **| J|IRA|** | :| As| mentioned|,| J|IRA| is| a| popular| At|las | sian|
suite|...
```

(b) Modified output token sequence

*Figure 6.* Responses to the prompt "`What is a good tool to plan a complex server deployment?`".

```
The| easiest| way| to| invest| in| property|…  Real| estate| investment|
trusts| or| RE|IT|s|,| real| estate| mutual| funds| may| be| the| easiest|.|…
There| are| many| options| for| acquiring| income| such| as| ground| level|
rental| or| owning| a| building| through| a| partnership|.| The| highest|
performing| investment| may| remain| a| gamble| and| have| no| guarantee|.|
The| next| hightest| would| have| to| be| investing| in| stocks| and| bonds|,
the| old| main|stay|.| Div|idend| and| bonds| have| higher| reliability|…
Note|:| the| previous| responses| and| answers| have| been| simplified|…
```

(a) True output token sequence

```
           (8)
The| eas | iest| way| to| invest| in| property|…  Real| estate| investment|
     (2)
trust | s| or| RE|IT|s|,| real| estate| mutual| funds| may| be| the| easiest|.|…
                             (6)
There| are| many| options| for| acqu | iring| income| such| as| ground|
                  (7)
level| rental| or| ow | ning| a| building| through| a| partnership|.| The|
                                       (3)
highest| performing| investment| may| remain| a| gam | ble| and| have| no|
guarantee|.| The| next| hightest| would| have| to| be| investing| in| stocks|
                        (4)            (1)
and| bonds|, the| old| main|st | ay|.| Div|id | end| and| bonds| have| higher|
     (2)
reli | ability|…  Note|:| the| previous| responses| and| answers| have| been|
     (5)
simpl | ified|…
```

(b) Modified output token sequence

*Figure 7.* Responses to the prompt "`What is currently the easiest investment opportunity with the capital and the highest game?`".

