# OpenReview forum: "Is Your LLM Overcharging You? Tokenization, Transparency, and Incentives"
_ICML.cc/2025/Workshop/TokShop — TokShop_

### Official Review · Reviewer_yvzJ · 2025-06-07
**An insightful and critical analysis of LLM-as-a-service pricing, highlighting vulnerabilities and proposing an elegant incentive-compatible solution.**

**Rating:** 9
**Confidence:** 4

**Review:**

**Summary of the Paper:**

This paper critically examines the prevalent pay-per-token pricing mechanism used by cloud-based large language model (LLM) services. It identifies a significant information asymmetry between the LLM provider and the user, where the provider observes the full generative process but the user only sees and pays for the final output tokens. This asymmetry creates a moral hazard, incentivizing providers to misreport the number of tokens to maximize their utility. The core issue is that the tokenization of a given string is not unique. For example, "San Diego" could be reported as two tokens or nine individual characters ("S|a|n| |D|i|e|g|o|"), leading to overcharging without changing the actual output text.
The authors formalize this interaction as a principal-agent problem and demonstrate that, under the current pay-per-token model, providers have a financial incentive to report the longest possible tokenization, potentially by reporting each character as a separate token, which could lead to overcharging users. While transparency about the LLM's generative process (e.g., next-token distribution) makes optimal misreporting (finding the longest plausible tokenization) NP-Hard, the paper introduces an efficient heuristic algorithm (Algorithm 1) that, as a proof-of-concept, allows providers to find plausible longer tokenizations. Experiments using models from the Llama, Gemma, and Ministral families show that this heuristic can lead to overcharging users by up to ~13% without raising suspicion, especially at higher temperature values commonly used in creative tasks.
To address this vulnerability, the paper proposes a novel incentive-compatible token pricing mechanism called pay-per-character. Under this mechanism, the price users pay is based on the number of characters in the output, completely eliminating the financial incentive for providers to misreport tokenizations. This is because an incentive-compatible mechanism ensures the monetary reward depends solely on the output string, not its tokenization. The authors suggest setting the price per character based on the average characters per token to minimize financial overhead during transition (e.g., rc = r0/cpt). Furthermore, they highlight that this mechanism creates positive incentives for providers to generate the shortest possible output token sequences, leading to beneficial downstream effects like reduced energy consumption, faster inference, and better context window utilization.

**Strengths:**

* Identifies a Critical Problem: The paper identifies a significant, yet overlooked, economic vulnerability in the widely adopted pay-per-token pricing model for LLM-as-a-service, which directly impacts user costs and provider incentives.

* Rigorous Theoretical Foundation: It effectively frames the issue as a principal-agent problem and provides strong theoretical results, including the NP-hardness of optimal misreporting under transparency requirements.

* Practical Proof-of-Concept: Despite the theoretical hardness, the introduction of an efficient heuristic algorithm (Algorithm 1) serves as a compelling proof-of-concept that demonstrates how providers can practically exploit the current pricing model without raising suspicion, showing real-world vulnerability.

* Clear and Provable Solution: The proposed pay-per-character pricing mechanism is simple, elegant, and theoretically proven to be incentive-compatible, completely eliminating the financial incentive for token misreporting.

* Positive Downstream Effects: The paper thoughtfully discusses how the pay-per-character model could positively incentivize providers to optimize tokenization algorithms for shorter sequences, leading to benefits like reduced energy consumption and faster inference, aligning provider incentives with broader efficiency goals.

* Empirical Validation: The study includes empirical experiments with diverse, state-of-the-art LLMs (Llama, Gemma, Ministral families) and prompts from a widely used dataset (LMSYS Chatbot Arena), lending credibility to the findings.

**Weaknesses:**

* Scope of Pricing Mechanisms: The work primarily focuses on additive pricing mechanisms. Further exploration of provider incentives under other pricing models, such as those based on output quality, could provide a more comprehensive understanding of the LLM-as-a-service economic landscape.

* Cost of Plausibility Evaluation: While Algorithm 1 is shown to be efficient, the assumption of negligible cost for plausibility evaluation by the provider is noted. More complex strategic algorithms might incur higher evaluation costs, which should be considered in a trade-off analysis.

* Implementation Challenges: The paper acknowledges that shifting from pay-per-token to pay-per-character would likely require external regulation or significant user pressure, highlighting a practical barrier to adoption despite the mechanism's theoretical benefits.

* Potential Misuse of Heuristic: The heuristic algorithm, though presented as a proof-of-concept, could potentially be misused by malicious providers. While the authors clarify their intent, this remains a concern, similar to adversarial attack research.

* Limited Scope of Strategic Behavior: The paper focuses on misreporting tokenization. As noted, providers could engage in other forms of strategic behavior, such as misreporting next-token distributions or even the output string itself, which are not explored in depth.

* Evaluation on Proprietary Models: While using open-weight LLMs is standard for research reproducibility, evaluating the potential for misreporting in widely used proprietary LLMs would offer more practical insights for the industry.

* Dataset Representativeness: The reliance on LMSYS Chatbot Arena prompts, despite its popularity, is acknowledged as potentially not fully representative of real-world user prompts, which could influence the generalizability of the empirical results.

---

### Official Review · Reviewer_bXJz · 2025-06-07
**Relevant and Timely Alternative to Pay-Per-Token Model**

**Rating:** 8
**Confidence:** 5

**Review:**

The paper investigates the economic and ethical issues with the popular pay-per-token pricing model that is used by major LLM providers today.  The key argument in the paper is that the model incentivizes the provider to misreport token utilization, thus potentially leading to overcharging and higher costs to the customers. The paper also demonstrates that it will be difficult for the customers to identify (or even suspect) when a provider is potentially misreporting tokens and overcharging, potentially up to 13%. As an alternative, the author proposes a pay-per-character model that takes away the ambiguity and also possibly aligns the incentives.

Strengths:
1. The topic is timely and underexplored at the intersection of machine learning, pricing economics, and customer usage of the model.
2. The principal-agent model and the hazard analysis frameworks are clearly articulated with supporting arguments/algorithmic evidence.
3. The author evaluates the deficiencies in the pay-per-token model and the proposed pay-per-character pricing model across multiple popular models available, such as Llama, Gemma, and Mistral.
4. The pay-per-character model is simple, significantly less ambiguous, and effective.

Weaknesses:
1. The author assumes that the providers now or in the future will be unwilling to share the details of the internal generative process. And as a result, the author does not push for increased transparency. But instead, proposes a solution based on the current levels of transparency by the providers.
2. The paper completely ignores possible manipulations with the pay-per-character model. One possibility is potentially increasing the character output, which eventually results in manipulation of the output and pricing.
3. The solution does not evaluate the overhead on providers to switch from pay-per-token to pay-per-character model. Depending on how fundamentally ingrained the pricing models are to "token" as the fundamental unit, as opposed to "character"

---

### Official Review · Reviewer_sQQn · 2025-06-09
**Timely study relevant to the workshop that mitigates an overlooked risk of API-based LLMs (tokenization misreporting for pricing inflation)**

**Rating:** 8
**Confidence:** 4

**Review:**

**Summary:**

This paper investigates a vulnerability in the pay-per-token pricing model used by API-based LLM providers. It shows that providers can inflate token counts by misreporting tokenization without altering the output string, thereby overcharging users. The authors frame this as a principal–agent problem, showing that optimal misreporting is NP-hard, but introduce a heuristic algorithm that enables overcharging of up to 13%. As a remedy, they propose a pay-per-character pricing model, which is incentive-compatible and removes the motivation to misreport tokens.

**Strengths:**
- The paper highlights a previously overlooked topic of financial misalignment in LLM pricing, framing the user-LLM provider interaction as a principal-agent problem
- Introduces a pay-per-character approach that could eliminate the financial incentive for misreporting tokenizations

**Weaknesses:**
- The paper assumes linear cost per token; real-world inference costs vary with model architecture and hardware utilization, so the linear cost assumption is a simplification
- The use of the term “transparent” is not clear. If a provider is transparent about the generative process, then it is not obvious how they could still overcharge without the user noticing. The paper should clarify what level or kind of transparency is assumed in its experiments, and whether that transparency is realistically verifiable or simply declared by the provider.

**Comments:**
- Lines 101-103: “Within this literature, the works by Cai et al. (Cai et al., 2025) and Saig et al. (Saig et al., 2024) are the most closely related to ours.” The sentence has a redundant and non-standard citation style, using \citet{} would fix the issue. Same applies for lines 113-117: “To reduce the financial incentive to strategize, Cai et al. (2025) argue for solutions based on increased transparency as well as trusted execution environments, and Saig et al. (2024) argue for a pay-for-performance pricing mechanism using a contract theory formulation.“
- It would be interesting to show how varying average characters-per-token (from different corpora or languages) affects the cost plan and revenue neutrality.

Overall, I believe this paper is timely and relevant to the workshop topics. I skimmed some of the theory as it is a non-archival submission, but I think it represents solid work and is worth presenting at the workshop as it highlights a real risk of API-based LLM solutions.

---

### Decision · Program_Chairs · 2025-06-10

Accept